# Quality of Life and Treatment Modalities in Patients with Interstitial Cystitis: The Patients’ Perspective

**DOI:** 10.3390/healthcare12040466

**Published:** 2024-02-13

**Authors:** Charlotte van Ginkel, Frank Martens, Mathilde Scholtes, John Heesakkers, Dick A. W. Janssen

**Affiliations:** 1Department of Urology, Radboudumc, 6525 GA Nijmegen, The Netherlands; charlotte.jvanginkel@radboudumc.nl (C.v.G.); frank.martens@radboudumc.nl (F.M.); 2Interstitial Cystitis Patient Association ICP, 4000 AB Tiel, The Netherlands; 3Department of Urology, Maastricht UMC+, 6229 HX Maastricht, The Netherlands; john.heesakkers@mumc.nl

**Keywords:** interstitial cystitis, bladder pain syndrome, quality of life, therapies, lifestyle

## Abstract

Background: Quality of life (QoL)-based outcomes are hardly incorporated into interstitial cystitis/bladder pain syndrome (IC/BPS) guidelines, because studies are limited and outdated. Therefore, guidelines might not reflect the current clinical situation accurately. Secondly, guidelines suggest using a multimodal approach for BPS/IC management, but data on the patient-perceived efficacy of these therapies are limited. The aim of this study is to investigate the perception of IC/BPS patients of their QoL, to determine which treatments they have received, and to examine how they evaluate the efficacy of these various (alternative) therapies. Methods: A quantitative retrospective database evaluation was performed, with data from an existing IC/BPS patient survey (*n* = 217) that was conducted in 2021. This survey contained QoL data based on validated questionnaires such as EQ-5D 5L. Results: The QoL of patients is affected significantly by IC/BPS. This is evident from the various affected domains on the EQ-5D 5L. The symptom severity was negatively affected by a delay in diagnosis, and there were clear differences in QoL domains between females and males. Secondly, coagulation therapy and intravesical glycosaminoglycan (GAG) therapy were most appreciated by patients. Other (alternative) treatments were commonly utilized, although some had doubtful results and high discontinuation rates. Conclusion: QoL is considerably impaired in IC/BPS patients. The diverse responses and adherence to various treatments warrant a personalized approach (phenotype-oriented therapy). To achieve QoL improvement, it is important to incorporate the patient’s perspective in treatment guidelines.

## 1. Introduction

Interstitial cystitis/bladder pain syndrome (IC/BPS) is defined by the International Society for the Study of Bladder Pain Syndrome (ESSIC) as a rare chronic inflammatory disease of the bladder, with symptoms of pain, urgency and/or frequency [1]. In 1995, Bade et al. conducted a study amongst urologists in the Netherlands and calculated the prevalence of IC/BPS to be around 8 to 16 per 100,000 female patients, with a female-to-male ratio of 10:1 [2]. This aligns with international literature [3]. There is an estimated 1860 IC/BPS patients diagnosed in the Netherlands, but it is likely there are many undiagnosed patients due to diagnostic delay and misdiagnosis [4]. The syndrome is uncommon, but it does severely impact the quality of life (QoL) of the individuals, resulting in a high burden on healthcare providers.

Although there are studies that report on QoL in IC/BPS [5,6,7,8,9,10,11,12,13,14,15,16], in the current guidelines, quality of life (QoL) is only sparsely mentioned. The European Association of Urology (EAU) and the Dutch Association of Urology do not elaborate on QoL [17,18]. The American Urology Association briefly discusses QoL and the adverse impact that IC/BPS has on patients [19]. These statements rely on literature dating back 15–20 years and do not incorporate improvements in healthcare management [5,7,9,10,11,12,13,14,15,16,20,21]. Evidence-based guidelines do not necessarily align with the patient’s perspective. Rare diseases often have few recommendations that have a sufficiently high level of supporting evidence, due to a lack of randomized controlled trials. This shows the importance of performing studies that also incorporate the patient’s perspective on treatment efficacy.

When treating IC/BPS, the EAU recommends adopting ‘multimodal approaches that encompass behavioral, physical and psychological techniques alongside the oral or invasive treatments’ [22]. While research has shed light on the efficacy and potential contributions of various therapies, it remains unclear how these therapies, including alternative approaches and behavioral modifications, are currently being utilized and perceived by patients in clinical practice.

Another gap in current knowledge pertains to the presentation of male IC/BPS patients. IC/BPS is a rare disease predominantly found in women, which has led to a research focus also primarily aimed at women. There is a limited understanding of the potential differences in symptoms, QoL or treatment efficacy between men and women.

The primary aim of this study is to evaluate how IC/BPS patients perceive their QoL in different domains and to investigate how patients rate the efficacy of different therapies offered to them, in order to obtain a better insight of the patients’ view on real life clinical practice. Additionally, the secondary objectives of this study are to scrutinize the data concerning demographics, symptomatology, therapies, sex differences and lifestyle adjustments related to IC/BPS.

## 2. Materials and Methods

### 2.1. Methodology and Ethical Statements

A quantitative database study was conducted to assess the effect of IC/BPS on patients’ QoL. The database used in this study was provided by the Dutch IC/BPS patient association (ICP) and consisted of survey data previously gathered by the ICP. Our institute approved this database under registration number: 115067. A review by the Ethical Review Board (Centrale Commissie Mensgebonden Onderzoek Oost Nederland) confirmed that no ethical consent application was needed because of the non-invasive and non-burdensome nature of this study (registration number 2023-16852).

### 2.2. Study Design

The database comprised responses to 147 questions that were created by a core group of IC/BPS patients and expert clinicians and were related to symptoms and QoL. The open access online survey was voluntarily completed by individuals affected by IC/BPS through the Dutch patients society for IC/BPS (ICP). The online survey was performed and collected in July 2021 using SurveyMonkey and the data were fully anonymized. The Dutch patient association ICP advertised this survey on their website and social media platforms. Data were stored electronically and securely and were only accessible by researchers. The database encompassed a diverse group, including those diagnosed with IC/BPS, former IC/BPS patients (who had undergone cystectomy) and partners. For the analysis, only adult patients who reported to have been diagnosed with IC/BPS and who reported to still have IC/BPS at the time of the questionnaire were included. Former patients and partners were excluded from this analysis.

The main outcome measures of this study were: the influence of IC/BPS on their daily lives, their current experiences of the disease and the EQ-5D 5L questionnaire on QoL. The secondary study outcomes were as follows: (1) patient characteristics, such as age, sex and subtype of IC/BPS, such as the presence of Hunner’s lesions (HLs), (2) symptomatology, (3) having been given or having used (alternative) treatments and their perceived efficacy by patients themselves and (4) applied lifestyle adjustments and their efficacy. These study outcome measures were collected from the database. The survey consisted of the following four parts: (1) The demographic section, wherein participants were asked about their sex and age category. (2) The diagnosis and symptom section, which included questions concerning the type of symptoms, the severity of symptoms, diagnosis, diagnosis delay, cystoscopy findings, urinary tract infection-related questions, the presence of concurrent chronic diseases and the validated O’Leary Sant Interstitial Cystitis Symptom Index and Problem Index (ICSI/PI) questionnaire for evaluating IC/BPS symptom presence and burden. (3) The quality-of-life section, in which, initially, participants were asked about the magnitude of the influence IC/BPS had on their daily lives, their current experiences of the disease, and any potential sleep-related challenges. The section ended with the administration of the EQ-5D 5L validated questionnaire, designed to assess their QoL in different domains. (4) The treatment section covered many different (alternative) treatment options for IC/BPS that are used by patients within the Netherlands. Participants were asked to answer, for each treatment, whether they had undergone the treatment, its efficacy on their symptoms, the presence of complications and their (dis)continuation with the treatment. The treatment options included medication, bladder instillations, neuromodulation, transurethral Hunner’s lesion (HL) resection, intravesical botulin injections, hydrodistension, pelvic floor physical therapy (PFPT) and specialized care in a pain center. Participants were also asked if they had considered undergoing bladder removal and the reasons behind their consideration. Additionally, the same questions used for standard clinical treatments were also applied to alternative therapies, such as cannabis oil, smoking cannabis, acupuncture, orthomolecular therapy, osteopathy, manual therapy and food supplements. Finally, a section of ten questions collected data concerning lifestyle adjustments, such as diet, meditation, exercise, stress management and sleep health.

### 2.3. Statistical Analysis

Data analysis was performed using SPSS v27 (Statistical Package for the Social Sciences, Chicago, IL, USA). Data are reported as numbers (%) with 95% confidential intervals, medians (interquartile range) or means (standard deviation). For questions with possible multiple answers, the percentages illustrate the percent of total cases reported. Where comparisons were made, the independent sample *t*-test was used for normally distributed data and the Mann–Whitney U test for not-normally distributed data. Correlations were assessed with (1) cross tabs, interpreting the linear-by-linear association or the Chi square and (2) linear regression analysis for continuous variables. For all tests, the significance was set at 5%.

## 3. Results

### 3.1. Demographics

In the analysis, 193 participants were included; the demographic data are summarized in Table 1A. The majority of patients were female (89%). The age distribution showed that 33% were between 41 and 60 years old, while 54% were between 61 and 80 years old. Over 60% of patients received their diagnosis within the last 10 years at the time of filling in the survey, with a smaller group receiving diagnosis >10 years ago. In almost all patients, diagnosis was made by a urologist with cystoscopy. In 46% of the patients, HLs were identified. Half of the patients were affected by concurrent urinary tract infections and symptoms of irritable bowel syndrome. Auto-immune associations, such as allergies, rheumatoid arthritis and fibromyalgia, were seen in 22–38% of the cases.

### 3.2. Symptoms

The data showed that there was a considerable diagnostic delay of >6 years in 30% of patients. They also showed that this diagnostic delay decreased over time, implying that there was an improvement in recognizing and diagnosing the syndrome by clinicians. The most frequently reported symptom was pain (92%) and was the top priority in treatment for over 60% of the patients. Frequency and urgency were commonly reported symptoms, but for urgency, up to 76% of the patients reported either no problem or only a minor issue. Nearly all patients experienced nocturia, with 50% needing to urinate four or more times during the night. Moreover, although 76% of the patients reported experiencing sleep difficulties, and over half of the study population experienced moderate to big sleep problems, the priority for addressing these symptoms in treatment was lower compared to other symptoms. The presence of pain leads to significantly more sleep problems (*p* = 0.007). A summary of the experienced symptoms and burden are shown in Table 1B. There were no significant differences in reported symptoms and severity between patients with (HL+) and without HLs (HL−).

### 3.3. Quality of Life

Data concerning QoL are shown in Table 2. About half of the patients in our cohort experienced symptoms at least two years before receiving their diagnosis. For a considerable number of patients, this diagnostic delay extended beyond ten years. A prolonged diagnostic delay led to a statistically significant increase in the experienced pain (*p* = 0.015), difficulties related to pain (*p* = 0.038) and difficulties concerning urinary frequency (*p* = 0.049).

More than 80% of the patients reported a moderate to severe impact of IC/BPS on their daily lives. Among HL+ patients, this impact was significantly greater than for those without HLs, with respective means of 4.75 versus 4.34 (range: 1–6, *p* = 0.019), but this difference was clinically very small. The majority of the patients (72%) reported that, at the time of the survey, they perceived that the condition was fading to the background or that it was acceptable in terms of living with the condition, while a smaller portion struggled or even felt overwhelmed by the disease. This did not correlate with how long someone had lived with IC/BPS.

Furthermore, the patients’ QoL was impacted across various domains, as assessed by the EQ-5D 5L questionnaire, divided in ‘any problems’ and ‘no problems’. The domains most notably affected were pain (93%) and usual activities (75%). Over half of the patients reported having feelings of anxiety and/or depression. Lastly, concerning self-care, the least-affected domain, nearly all patients stated they were having no problems in this area.

There was a significant correlation between the experience of the disease and the impact on the domain ‘pain’ (R^2^ = 0.55; *p* < 0.001), with patients who scored worse on this domain also experiencing their disease as more severe. This correlation was also found for the anxiety domain (R^2^ = 0.018; *p* = 0.006). Twenty seven percent of the participants said they had considered bladder removal, with the three most occurring reasons being unbearable pain (41%), a low QoL (29%) and no other treatment options (14%).

### 3.4. Treatments

#### 3.4.1. Standard Treatments

All patients received treatment from a urologist, while a gynecologist was also involved in the treatment in 50% of the cases. Pain specialists were only engaged in the care of 17% of the patients. A similar percentage had been referred for counseling from a psychologist/psychiatrist for their IC/BPS complains. Patients who experienced more feelings of anxiety sought counseling from a psychologist/psychiatrist more often (*p* = 0.014).

Concerning treatment, in Table 3, the summary of all reported treatments is shown. A wide range of therapies is available, and the most commonly offered ones include bladder instillations (although it is undefined as to which ones) (81%), pelvic floor physical therapy (PFPT) (77%), medications (75%), for example paracetamol (67%), tricyclic antidepressant (23%) and nonsteroidal anti-inflammatory drug (NSAIDs) (21%), neuromodulation (39%), for example percutaneous tibial nerve stimulation (PTNS) (47%), transcutaneous electrical nerve stimulation (TENS) (39%) or sacral nerve stimulation (SNS) (15%) and, lastly, transurethral HL coagulation (30%). Of the HL+ patients, 60% underwent transurethral cystoscopic coagulation of the HL. There was no difference in the use of bladder instillations between HL+ and HL− patients.

#### 3.4.2. Alternative Treatments

When examining the efficacy of these therapies on symptoms as reported by patients themselves, transurethral HL coagulation and bladder instillations (undefined) yielded the best results. For pain, these treatments showed improvements in, respectively, 76% and 65% of the patients. Although only 17% of patients had received treatment in a specialized pain treatment center, they reported improvement in pain symptoms in more than half of the cases. For urgency and frequency, the success rate was lower. Transurethral coagulation of HL was the only treatment that demonstrated notable improvement in sleep difficulties, with approximately one-third of the patients experiencing positive outcomes.

Intravesical botulin A injections were given to 20% of the patients; in that subgroup, those patients reported significantly higher urinary frequency (90% vs. 75%, *p* = 0.041) and urgency (85% vs. 79%, *p* = 0.045). There was a slight correlation observed in these patients between the desire to treat urgency and the use of intravesical botulin A injections (*p* = 0.058).

Unfortunately, numerous patients did not experience any beneficial effect from various treatments. For PFPT, neuromodulation (especially PTNS and TENS) and hydrodistension, over half of the patients reported no positive impact. A noticeable number of patients discontinued these treatments, with the majority (84%) of those having received PFPT reported that they had stopped. For transurethral HL resection and bladder instillations, these numbers were evidently better. This demonstrates that treatment response varied between patients.

Regarding medication (Table 3C), more than two-third of the patients reported the use of paracetamol, which had a positive effect on symptoms in half of the cases. Other drugs, such as tricyclic antidepressants (like amitriptyline), NSAIDs and tramadol, were used in fewer patients, and they also had less impact in alleviating symptoms. Oxycodone was rarely used and had a minimal effect on symptoms. Half of the patients reported the use of prophylactic antibiotics over the past years for urinary tract infections.

In addition to standard care, numerous patients (73%) sought relief through alternative therapies (Table 3B). Among these, food supplements (such as Probiotics, Cystoprotek^®^, Prelief^®^ and turmeric) were the most commonly used. Combined, they had the highest success rate, with beneficial effects reported by81% of the patients using them. However, from the group that used food supplements, only probiotics were frequently used and considered effective in 36% of these patients, while all the other supplements were used less frequently by patients. Other alternative therapies, such as acupuncture, cannabis oil and osteopathy, were used as well. However, a large percentage of patients reported no positive impact on symptoms of these therapies, which led to a high number of patients who discontinued them.

#### 3.4.3. Lifestyle Adjustments

Adjustments in lifestyle were frequently adopted by patients (Table 4). Of the patients, 66% reported changing their diet. Moreover, the majority of the patients adapted their lifestyle on different levels as well, such as stress management (62%), exercise (38%), meditation (24%) and/or sleep health (22%). The urologist played a primary role in providing dietary advice, while the patient association also contributed to informing the patients about these changes. In terms of other lifestyle adjustments, the additional contribution of the internet and personal experiences played a role.

Among the various lifestyle modifications, dietary changes demonstrated the most positive impact on symptoms, particularly in relation to pain relief. A small percentage of patients (22%) reported no noticeable effect, and an even smaller portion stopped their dietary changes. The second most applied and beneficial lifestyle adjustment was stress management, with a similar ratio of patients reporting no effect or discontinuing this change. Exercise and meditation yielded slightly lower success rates overall, except for sleep problems. Sleep-related lifestyle changes had a significant positive impact on sleep problems but had comparatively less effect on other symptoms.

### 3.5. Men Versus Women

This study cohort included 20 male patients, which is of interest since they are rarely evaluated as a separate group for analysis in IC/BPS (Table 5). The male patients were slightly older; 85% were older than 60 years, in comparison to 55% of women (Table 5A). There were no significant differences in the type of IC/BPS, but the HL subtype was slightly more reported (55% versus 45%). Looking at symptoms, male patients presented with more lower urinary tract symptoms (LUTS), significantly more frequency complaints (*p* = 0.013) and a trend towards more urgency symptoms (*p* = 0.065) (Table 5B).

Concerning QoL, the influence of IC/BPS on male patients’ lives and their experiences of the disease did not differ from the female population (Table 5C). However, male patients did score significantly higher on the self-rated health score (EQ VAS, mean 66.2 ± 19.9 versus 54.1 ± 25.9 (*p* = 0.045)) and had significantly fewer problems with anxiety (25%) in comparison to women (57%) (*p* = 0.007).

For treatments, there was only a significant difference for transurethral HL coagulation (Table 5D). Zooming in on the HL subgroup, male patients (*n* = 11) had undergone significantly higher rates of transurethral HL coagulation in comparison to female HL+ patients (*n* = 70), *p* = 0.027. The beneficial effect of this therapy on urinary frequency was better in the male population (80% reported a positive effect versus 47% of women, *p* = 0.058). Bladder instillations were also used slightly more often in men, 90% versus 79%, but this was not significant. Male patients made fewer use of alternative therapies, 40% versus 73% (*p* = 0.023). Lifestyle adjustments were applied in similar amounts, 75% versus 77%.

## 4. Discussion

This study showed that the reported QoL of this IC/BPS cohort of patients was impaired, with a reported mean score of fifty-five out of one hundred percent for general QoL. This is also evident from the self-reported impact by patients in the various affected QoL domains (in particular pain, everyday activities and anxiety-related issues), as well as the delayed diagnosis, with the latter being >6 years for 30% of participants. We did not include a comparative group with heathy patients in this study. The data reported by patients showed that the longer the diagnostic delay was present, the worse the IC/BPS symptoms were. Pain was the dominant symptom and, together with anxiety, these symptoms affected subjective perceptions of disease severity. Interestingly, in our study, there were fewer reported issues with and less emphasis on addressing urgency, a key symptom of IC/BPS. Another important finding was the occurrence of problems with sleep. Finally, the third major finding was the wide array of (alternative) treatments utilized, although with doubtful results and high dropout rates.

Regarding the primary question of this study, our findings are in line with previous research that reported that the IC/BPS population experiences lower health-related QoL, compared to healthy age-matched controls [5,8,10,13]. This study revealed that HL+ patients experience a significantly greater impact on their lives compared to HL− patients. Even though this difference was clinically very small and the symptom burden did not differ within our study, this small difference still underlines that these patients could be regarded as a more severe subgroup [19]. For all patients, there was a more positive disease experience in their current situation than what might have been expected based on their current symptoms and their impact on QoL domains. This could suggest that these patients have adopted certain coping mechanisms. Previous research has linked catastrophizing and dwelling on distress to poorer experiences of the disease and prolonged distress [9,23,24]. Conversely, seeking emotional or social support has been associated with better adjustment to coping with the syndrome [25,26]. Survey bias can lead to both the underreporting or overreporting of symptom severity by patients. The study population is drawn from the patient association, which could have an undetermined effect on symptom reporting.

The second objective of this study covered demographics and symptoms. The study population resembled the known IC/BPS population concerning sex and age distribution [5,15]. A diagnostic delay, as highlighted in previous research, was also negatively correlated with pain in our own study [5]. Not surprisingly, together with anxiety, pain is associated with a more negative perspective on the disease. The process of pain sensitization has been well-described for chronic pain patients [27]. Nickel et al. found comparable results, and reported that depression scores were significantly correlated with a poorer QoL and increased pain [10]. Furthermore, mental health disorders are common in patients with urological pain syndromes [11]. Even though urgency was not reported as a very dominant symptom in our survey, it is a key symptom of IC/BPS. The sensation of urinary urgency often differs in IC/BPS, compared to patients suffering from overactive bladder (OAB), and relates more to pain and discomfort during bladder filling, instead of the sudden onset urgency with a fear of incontinence that is more typical for OAB [28]. Our study also revealed a high prevalence of sleep disturbances. These results are consistent with previous research [6,15]. As stated by Nickel et al., these disturbances can encompass more than just nocturia and potentially set up a vicious cycle, as poor sleep itself is linked to mood disturbances and diminished QoL [15,29,30,31]. This study demonstrated that patients do not prioritize addressing sleep problems in their treatment, but this may also be because this topic receives insufficient attention by physicians.

A third focus was the patient’s perceived efficacy of various treatments. Bladder instillations were received most often and were highly appreciated by patients. In the Dutch clinical guidelines, these bladder instillations mostly concern glycosaminoglycan (GAG) instillations, with a shift from dimethyl sulfoxide to GAG instillations occurring in 1995 [2,18]. European guidelines recommend intravesical GAG therapies with a low level of evidence, but it is not mentioned in the AUA guidelines, since it is not approved by the U.S. Food and Drug Administration (FDA). The latter still mainly focuses on dimethyl sulfoxide [18,19,22]. Transurethral resection of HL is strongly recommended in all guidelines and is commonly received with good success rates in our study, indicating its suitability for HL+ patients. Availability and treatment approaches vary throughout the world. This is reflected in our study, as there are no triamcinolone injections available in the Netherlands and a low use of pentosan polysulfate, which only became available and reimbursed in 2020 within the Netherlands. Also, there are no data on oral cyclosporine, which only became available off-label in 2023.

This study highlighted a concerning number of patients who experienced no therapy benefit and who discontinued treatments. This emphasizes the importance of personalized care, timed evaluations of treatment efficacy and stratifying patients, to determine the most suitable treatment for each individual. How and when therapy is applied could be of importance. In case of PFPT, only 40% received biofeedback and 17% received trigger point therapy. Our study was not designed to specifically measure therapy efficacy but reflects the patient experience and their adherence to treatment. The data were unsuitable to evaluate whether treatments were combined, such as PFPT with pharmacological pain management, which could influence the patient-perceived efficacy. Therapies in the alternative field were often explored by patients themselves but yielded only minimal beneficial effects on symptoms. This does signal a need for symptom improvement beyond what current standard clinical treatments are providing.

Guidelines acknowledge the additional significance of behavioral modification therapies. Our study shows that patients have successfully adopted this. In particular, dietary adjustments and stress management yielded the best results in our study, and these are also recommended by guidelines. We observed that patients were able to adhere to lifestyle modifications when implemented. It is worth noting that our study did not measure whether the level of behavioral modifications were effective in the long-term, but the observed effects seem substantial enough for patients to continue implementing these changes for their benefit.

Lastly, the analysis of the male subgroup of IC/BPS revealed a 1:9 male to female ratio in this dataset and an older age distribution compared to women. This is largely consistent with findings of Tincello et al. [5]. In our cohort, male patients presented with more LUTS complaints, which makes them a more challenging group to identify within an already rare disease. HL− male patients could be under- or misdiagnosed because of potential prostate-related diseases that are prevalent in the reported age category. Male patients rated their overall health higher compared to women, and also reported significantly fewer issues with anxiety. Serious problems with mental health were reported at similar rates for male and female IC/BPS patients [11]. It can be speculated that male patients both received and preferred more thorough treatment options, as they underwent transurethral coagulation of HL significantly more often and sought alternative therapies less frequently.

There are some limitations to this study. One potential limitation of this study lies in its retrospective nature, which could potentially increase the likelihood of recall bias. No comparator group of age-matched healthy volunteers was used in this study and the male group was still comparatively small. The study was conducted with patient data from one single Western European country, and the healthcare management strategies for IC/BPS patients can differ between countries. Patients provided information concerning their diagnosis themselves, and these data could not be confirmed using actual patient clinical data, due to the applied study methodology. This has to be taken into account when interpreting the reported differences in symptom severity between HL patients and non-HL patients.

For patient-reported satisfaction with treatments, there were limitations as to how to interpret these. Not all therapies can be directly compared to each other, since they may have been offered to patients at different stages with periods of different symptom severity/burden. Also, the perception of treatment efficacy could be mediated by other treatments that were given at the same time. The survey was made accessible to individuals through the Dutch IC/BPS patient association. Nonetheless, since participation was voluntary, it may be susceptible to response bias. This may be reflected in the relatively high number of HL patients in this dataset. Questions were designed to limit survey bias; however, it is important to note that only a minor fraction of the questions (approximately 10%) were derived from validated questionnaires. Together, these factors could potentially lead to an underrepresentation of IC/BPS patients experiencing mild symptoms.

## 5. Conclusions

This study reaffirms that the QoL of individuals with IC/BPS is impaired. Current practice in the Netherlands shows a reduction in diagnostic delay over time, which signals an improved awareness by clinicians. Treatments like cystoscopic coagulation therapy and intravesical treatments are most appreciated by patients. The diverse responses and sometimes poor efficacy and adherence to various treatments offered is an intriguing issue, which could possibly be addressed by a more personalized approach. Phenotype-oriented therapy could potentially yield better outcomes. Clinicians should take into consideration the fact that there are sex differences in BPS/IC symptom burden and treatment appreciation. The ongoing challenge is to underscore the importance of incorporating QoL and patient perspectives of treatments into disease management guidelines.

## Figures and Tables

**Table 1 healthcare-12-00466-t001:** Demographic overview and symptoms. This table presents data on sex, age, IC/BPS subtypes and comorbidities, along with information on main symptoms, symptom scores and the burden and/or prioritization of treatments.

	**All Patients (*n* = 193)**		
**A. Demographics**	**N (%)**		
Sex			
Female	166 (89)		
Male	20 (11)		
Age			
19–40 years	16 (9)		
41–60 years	62 (33)		
>60 years	109 (58)		
Subtype IC/BPS ^1^			
Type 1 (normal bladder wall)	6 (3)		
Type 2 (glomerulations)	81 (44)		
Type 3 (Hunner’s lesions)	84 (46)		
Unknown	13 (7)		
Comorbidities			
Bacterial cystitides	104 (54)		
Irritable bowel syndrome	56 (45)		
Allergy or sensitivity	47 (38)		
Rheumatoid arthritis	28 (22)		
Fibromyalgia	28 (22)		
Migraines	27 (22)		
**B. Symptoms**	**N (%)**		**N (%)**
Main symptoms ^2^			First priority in therapy ^3^
Pain	177 (92)		107 (56)
Frequency	151 (78)		37 (19)
Sleep difficulties	147 (76)		26 (14)
Urgency	140 (73)		20 (11)
	**Mean ± SD**	**Range**	
**O’Leary Sant Questionnaire ^4^**			
IC Symptom Index			
Total score	12.9 ± 4.6	0–20	
Urgency	3.24 ± 1.6	0–5	
Frequency	3.37 ± 1.6	0–5	
Nocturia	3.24 ± 1.5	0–5	
Bladder pain	3.02 ± 1.4	0–5	
IC Problem Index			
Total score	8.9 ± 3.7	0–16	
Urgency	1.30 ± 1.3	0–4	
Frequency	2.40 ± 1.2	0–4	
Nocturia	2.44 ± 1.4	0–4	
Bladder pain	2.72 ± 1.2	0–4	
Do you experience any problems with sleep?			
0 (no problem)–5 (big problem)	2.40 ± 1.5	0–5	

^1^ Based on the ESSIC classification of types of bladder pain syndrome. ^2^ Percent of cases—more than one can be present in each individual. ^3^ Percentages reported are from patients who filled this question in (which was less than under ^2^). ^4^ The O’Leary ICSI/PI questionnaire consists of the four-item ICSI (symptom index) that measures severity of daytime urgency frequency, nighttime urination and pain or burning, and a four-item ICPI (problem index) concerning how problematic symptoms are perceived. A scoring system between 0 and 20 is used. A total score (ICSI + ICPI score) of >6 suggests that IC/BPS is possible and a score > 12 is strong evidence in favor of a diagnosis of IC/BPS.

**Table 2 healthcare-12-00466-t002:** Impact of symptoms of IC/BPS upon quality of life. This table presents the perceived burden on quality of life by IC/BPS, as well as the results from the validated questionnaire EQ-5D 5L.

	**All Patients (*n* = 193)**	
	**N (%)**	
Symptoms before diagnosis		
0–2 years	88 (46)	
3–5 years	45 (23)	
6–10 years	17 (9)	
11–20 years	23 (12)	
>20 years	17 (9)	
Unknown	3 (2)	
	**Mean/Median**	**SD/IQR**
What is the influence of IC/BPS on your life? 0 (very small)–5 (very big)	3.54	1.18
How do you experience your IC/BPS at this moment? 0 (no complaints)–4 (overwhelmed)	2.13	0.98
**EQ-5D-5L (QoL)**	**N (%)**	
Mobility		
No problems	88 (46)	
Any problems	104 (54)	
Self-Care		
No problems	170 (88)	
Any problems	23 (12)	
Usual Activities		
No problems	48 (25)	
Any problems	145 (75)	
Pain		
No problems	14 (7)	
Any problems	178 (93)	
Anxiety		
No problems	91 (47)	
Any problems	102 (53)	
EQ VAS scale ^1^ (0–100)		
Mean ± SD	55.9 ± 25.3	

^1^ The validated questionnaire EQ-5D 5L assesses patient-reported QoL (impairments) using an overall score (the EQ visual analogue scale (VAS) runs from 0–100, with 100% being an optimal QoL) for QoL in different subdomains such as mobility, pain, anxiety, etc.

**Table 3 healthcare-12-00466-t003:** Reported therapies received and perceived efficacy This table presents data regarding the reported (alternative) therapies and their effects on symptoms and adherence to the treatment.

	**N (%)**	**N (%)**	**N (%)**	**N (%)**	**N (%)**	**N (%)**	**N (%)**
**A. Therapies**	**Received Treatment**	**No Effect**	**Pain**	**Urgency**	**Frequency**	**Sleep**	**Stopped Treatment**
Bladder instillation	153 (81)	46 (30)	100 (65)	61 (40)	55 (36)	24 (16)	86 (56)
Pelvic floor physical therapy	146 (77)	84 (58)	38 (26)	29 (20)	29 (20)	7 (5)	124 (84)
Neuromodulation	75 (39)	42 (57)	24 (32)	16 (22)	12 (16)	4 (5)	48 (67)
Laser/coagulation	58 (30)	12 (21)	44 (76)	33 (57)	32 (55)	21 (36)	22 (42)
Hydrodistension	58 (30)	33 (56)	18 (31)	19 (32)	17 (29)	10 (17)	-
Intravesical botulin	40 (21)	19 (48)	14 (35)	10 (25)	11 (28)	4 (10)	25 (63)
Therapy in a pain center	31 (16)	12 (39)	18 (58)	4 (13)	3 (10)	3 (10)	19 (63)
**B. Alternative Therapies**	**Received Treatment**	**No Effect**	**Effect on Pain**	**Effect on Urgency**	**Effect on Frequency**	**Effect on Sleep**	**Stopped Treatment**
Food supplements ^2^	73 (38)	14 (19)	-	-	-	-	33 (46)
Acupuncture	59 (31)	36 (60)	22 (37)	8 (13)	8 (13)	5 (8)	50 (85)
Cannabis oil	54 (28)	28 (52)	18 (33)	4 (7)	4 (7)	17 (31)	38 (71)
Osteopathy	44 (23)	29 (64)	14 (31)	7 (16)	7 (16)	6 (13)	31 (72)
Orthomolecular therapy	28 (15)	13 (46)	13 (46)	4 (14)	5 (18)	5 (18)	12 (50)
Manual therapy	11 (6)	7 (58)	5 (42)	2 (17)	0 (0)	1 (8)	7 (64)
Smoking cannabis	10 (5)	1 (9)	8 (72)	1 (9)	1 (9)	6 (55)	9 (69)
**C. Medication**	**Current Use**	**Positive Effect on Symptoms ^1,3^**					
	**N (%)**	**N (%)**					
Paracetamol	96 (67)	65 (51)					
Amitriptyline	33 (23)	22 (17)					
NSAIDs	30 (21)	33 (26)					
Tramadol	26 (18)	11 (9)					
Gabapentin/pregabalin	14 (10)	10 (8)					
Pentosan polysulfate	14 (10)	28 (22)					
Oxycodone	11 (8)	14 (11)					

Percentages are in percent of cases; ^1^ this includes all patients, also earlier use; ^2^ this includes: Cystoprotek^®^ (3), Prelief^®^ (7), probiotics (45), turmeric (22) and others (36); ^3^ effect on symptoms, answered yes or no. % shown are percentage of patients with positive response.

**Table 4 healthcare-12-00466-t004:** **Reported lifestyle adjustments.** This table presents the reported lifestyle adjustments and patients’ adherence to them. Moreover, the effects on symptoms are shown.

	**N (%)**	**N (%)**	**N (%)**	**N (%)**	
**Lifestyle Adjustments**	**Applied Adjustments**	**No Effect**	**Stopped**		
Dietary	127 (66)	24 (22)	17 (16)		
Avoid stress	87 (62)	16 (20)	15 (18)		
Exercise	54 (38)	15 (31)	4 (8)		
Meditation	34 (24)	9 (28)	3 (9)		
Sleep	31 (22)	10 (33)	4 (13)		
	**Effect on Pain**	**Effect on Urgency**	**Effect on Frequency**	**Effect on Sleep**	
Dietary	78 (73)	34 (32)	28 (26)	18 (17)	
Avoid stress	37 (46)	9 (11)	10 (13)	8 (10)	
Exercise	15 (31)	6 (12)	4 (8)	9 (18)	
Meditation	13 (41)	1 (3)	1 (3)	8 (25)	
Sleep	3 (10)	0 (0)	2 (7)	15 (50)	

Percentages are the percent of cases.

**Table 5 healthcare-12-00466-t005:** Differences between male and female participants. This table summarizes the most important findings between male and female participants, including demographics, symptoms, quality of life and treatments.

	**Male (*n* = 20)**		**Female (*n* = 166)**		
**A. Demographics**	**N (%)**	**(95% CI)**	**N (%)**	**(95% CI)**	***p*-Values**
Age					*p* = 0.082
19–40 years	1 (5)	(1–25)	15 (9)	(5–14)	
41–60 years	2 (10)	(1–32)	60 (36)	(29–44)	
>60 years	17 (85)	(62–97)	91 (55)	(47–63)	
Subtype IC/BPS ^1^					*p* = 0.770
Type 1 (normal bladder wall)	2 (10)	(2–32)	3 (2)	(0–5)	
Type 2 (glomerulations)	6 (30)	(12–54)	72 (46)	(38–54)	
Type 3 (Hunner’s lesions)	11 (55)	(32–77)	71 (45)	(37–53)	
**B. Symptoms**					
Main symptoms ^2^					
Pain	17 (85)	(62–97)	153 (94)	(87–96)	*p* = 0.281
Frequency	20 (100)	(83–100)	126 (77)	(69–82)	*p* = 0.013
Sleep difficulties	15 (75)	(51–91)	127 (78)	(69–83)	*p* = 0.881
Urgency	18 (90)	(68–99)	117 (72)	(63–77)	*p* = 0.065
**C. Quality of Life**	**Mean**	**SD**	**Mean**	**SD**	***p*-Values**
What is the influence of IC/BPS on your life?					
0 (very small)–5 (very big)	4.55	1.91	4.53	1.18	*p* = 0.927
How do you experience your IC/BPS at this moment?					
0 (no complaints)–4 (overwhelmed)	3.21	0.92	3.11	0.99	*p*= 0.927
**EQ-5D 5L**	**N (%)**	**(95% CI)**	**N (%)**	**(95% CI)**	***p*-Values**
Mobility					*p* = 0.778
No problems	9 (47)	(24–71)	73 (44)	(36–52)	
Any problems	10 (53)	(29–76)	93 (56)	(48–64)	
Self-Care					*p* = 0.789
No problems	18 (90)	(68–99)	146 (88)	(82–92)	
Any problems	2 (10)	(1–32)	20 (12)	(8–18)	
Usual Activities					*p* = 0.521
No problems	6 (30)	(12–54)	39 (24)	(17–31)	
Any problems	14 (70)	(46–88)	127 (76)	(69–83)	
Pain					*p* = 0.177
No problems	0 (0)	(0–17)	14 (8)	(5–14)	
Any problems	20 (100)	(83–100)	151 (92)	(86–95)	
Anxiety					*p* = 0.007
No problems	15 (75)	(51–91)	72 (43)	(36–51)	
Any problems	5 (25)	(9–49)	94 (57)	(49–64)	
	**Mean**	**SD**	**Mean**	**SD**	***p*-Values**
EQ VAS scale (0–100)	66.15	19.87	54.05	25.85	*p* = 0.045
**D. Therapies**	**N (%)**	**(95% CI)**	**N (%)**	**(95% CI)**	***p*-Values**
Received Therapies ^2^					
Bladder instillations	17 (90)	(67–99)	129 (79)	(72–85)	*p* = 0.268
Pelvic floor physical therapy	15 (75)	(51–91)	126 (77)	(70–83)	*p* = 0.817
Neuromodulation	11 (55)	(32–77)	60 (36)	(29–44)	*p* = 0.101
Laser/coagulation	10 (50)	(27–73)	47 (29)	(22–36)	*p* = 0.051
Hydrodistension	3 (15)	(3–38)	54 (33)	(26–41)	*p* = 0.103
Intravesical botulin A	3 (15)	(3–38)	34 (21)	(15–27)	*p* = 0.555
Therapy in a pain center	1 (5)	(1–26)	29 (18)	(12–24)	*p* = 0.167
Alternative Therapies					
Yes	8 (40)	(19–64)	102 (68)	(59–75)	*p* = 0.023
No	12 (60)	(36–81)	49 (32)	(25–41)	
Lifestyle Adjustments					
Yes	15 (75)	(51–91)	125 (77)	(69–83)	*p* = 0.867
No	5 (25)	(9–49)	38 (23)	(17–31)	

Percentages represent the percent of cases; underscore means significant difference (*p*-value); ^1^ based on the ESSIC classification of types of bladder pain syndrome; ^2^ counts of participants who reported having received the treatment.

## Data Availability

The data presented in this study are available on request from the corresponding author. The data are not publicly available due to the fact that this was not specifically consented to by the ICP beforehand.

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
