# Peer review of "Quality of Life and Treatment Modalities in Patients with Interstitial Cystitis: The Patients’ Perspective"

_healthcare, 2024, doi:10.3390/healthcare12040466_

Round 1

Reviewer 1 Report

Comments and Suggestions for Authors

General comment:

The present manuscript is a cross-sectional study that evaluates the perception of IC/BPS patients on their QoL and to determine which treatments they received and how they evaluate the efficacy of these treatment. Although, in my opinion, the topic of the manuscript is up-to-date, and the results relevant to clinical practice, several points should be improved.

Specific comments:

Abstract:

The authors employed abbreviations without providing explanations. Since the abstract may be read independently, clarity can be enhanced by explaining these abbreviations.

Introduction

·       Line 32: the abbreviation "ESSIC" is used without clarification.

·  Lines 39-40: the statement "it does impact a number of patients in the Netherlands" is unclear. Please rephrase for improved clarity.

     Material and Methods

In lines 80-81, the authors mentioned including only patients with the IC/BPS diagnosis at the time of the questionnaires. Is there information on the mean/median time between diagnosis and study inclusion?

Additional information about the EQ-5D 5L questionnaire used to evaluate quality of life and the O’Leary Sant Interstitial Cystitis Symptom Index and Problem Index (ICSI/PI) would enhance comprehension of the results, especially in tables.

Results

·       The results section was overly exhaustive, affecting readability. Consider rephrasing for better clarity.

·       Table 1: I it is suggested to exclude information regarding "first priority in therapy" and incorporate it into the text. Clarify if participants could choose more than one option, as the percentages seem to sum to more than 100%. The information regarding 95%CI for the percentage, was in my opinion accessory and not add much information to the results, adding some noise to the table. I suggest removing this information from the table (this suggestion applies to all tables). Also, please improve the clarity of footnote number 2 in Table 1.

Discussion

·       The cross-sectional design of the study and the absence of a control group precluded the evaluation of the effect/impact of the IC/BPS on QoL. Therefore, the authors should have some precaution in the way some statements are done. For instance, the phrase “This study showed that the QoL is clearly affected by IC/BPS” was not compatible with the authors study design.

·       The limitations section needs to be expanded to include other important limitations, such as the cross-sectional design precluding the evaluation of causality, the small number of men precluding multivariate analysis, and others.

Comments on the Quality of English Language

The article should be reviewed for editing of English language

Author Response

General comment:

The present manuscript is a cross-sectional study that evaluates the perception of IC/BPS patients on their QoL and to determine which treatments they received and how they evaluate the efficacy of these treatment. Although, in my opinion, the topic of the manuscript is up-to-date, and the results relevant to clinical practice, several points should be improved.

Reply: Thank you for reviewing this manuscript and providing us with the relevant comments. Below we will answer your quarries. 

Specific comments:

Abstract:

The authors employed abbreviations without providing explanations. Since the abstract may be read independently, clarity can be enhanced by explaining these abbreviations.

Reply: thanks for your comments. We written in full all abbreviations in the abstract and the main text.

Introduction

  • Line 32: the abbreviation "ESSIC" is used without clarification. :

Reply: we have clarified this further in the text.; Interstitial cystitis/bladder pain syndrome (IC/BPS) is defined by the International Society for the Study of Bladder Pain Syndrome (ESSIC) as a rare chronic inflammatory disease of the bladder, with complaints of pain, urgency and/or frequency

  • Lines 39-40: the statement "it does impact a number of patients in the Netherlands" is unclear. Please rephrase for improved clarity.

Reply: We changes the sentence into: ‘The syndrome is uncommon, but it does severely impact quality of life (QoL) of the in-dividuals resulting in a high burden on healthcare providers.

     Material and Methods

In lines 80-81, the authors mentioned including only patients with the IC/BPS diagnosis at the time of the questionnaires. Is there information on the mean/median time between diagnosis and study inclusion?

Reply: we do have data on how long patients have had the disease categorized by 5yr intervals, but this is not sufficient to calculate a reliable mean/average

Additional information about the EQ-5D 5L questionnaire used to evaluate quality of life and the O’Leary Sant Interstitial Cystitis Symptom Index and Problem Index (ICSI/PI) would enhance comprehension of the results, especially in tables.

Reply: we have elaborated on the interpretation and meaning of EQ-5D-5L and ICSI/PI in the legends of the designated tables and text.

Results

  • The results section was overly exhaustive, affecting readability. Consider rephrasing for better clarity.

Reply: we struggled to summarize the data as much as possible with the multitude of outcomes evaluated. We rephrased and reorganized much of the text to make it more readable.

  • Table 1: I it is suggested to exclude information regarding "first priority in therapy" and incorporate it into the text. Clarify if participants could choose more than one option, as the percentages seem to sum to more than 100%. The information regarding 95%CI for the percentage, was in my opinion accessory and not add much information to the results, adding some noise to the table. I suggest removing this information from the table (this suggestion applies to all tables). Also, please improve the clarity of footnote number 2 in Table 1.

Reply: thanks for this comment : We removed the 95% CI information in all tables where it was indeed redundant. We improved legends and footnotes of Table 1 and clarified and corrected the percentages of ‘first priority in therapy’ as so it is more clear for readers. We do believe ‘first priority in therapy’ is important to assess and report in the table. Should there be a wish to still move it into text despite corrections, we will do so , naturally.  

Discussion

  • The cross-sectional design of the study and the absence of a control group precluded the evaluation of the effect/impact of the IC/BPS on QoL. Therefore, the authors should have some precaution in the way some statements are done. For instance, the phrase “This study showed that the QoL is clearly affected by IC/BPS” was not compatible with the authors study design.

Reply: thank you for this comment , we rephased the sentence and legends of relevant tables and elaborated more on the ED-5Q 5L questionnaire and ICSI/PI and how it to be interpreted

  • The limitations section needs to be expanded to include other important limitations, such as the cross-sectional design precluding the evaluation of causality, the small number of men precluding multivariate analysis, and others.

Reply: we added more information on study limitations based on your suggestions:

‘There are some limitations to this study. One potential limitation of this study lies in its retrospective nature, which could potentially increase the likelihood of recall bias. No comparator group of age matched healthy volunteers was used in this study and the male group was still comparatively small. The study was conducted with patient data from one single Western European country and healthcare management for IC/BPS patients can differ between countries. For patient reported satisfaction with treatment, there were limitations as to how to interpret these. Not all therapies can be directly compared to each other since they may have been offered to patients at different stages with periods of different symptom severity / burden. Also the perception of treatment efficacy could be mediated by other treatments that were given at the same time.’

Comments on the Quality of English Language. The article should be reviewed for editing of English language

Reply: We reviewed the entire manuscript for English writing

Reviewer 2 Report

Comments and Suggestions for Authors

Dear authors,

This is an excellent article. I have only some minor comments before it could be published.

- although the recruitment has been pooled by a national database, it remains unclear if the diagnosis of IC/BPS really meets the ESSIC criteria. You probably have to document the limitations occurring due to this common handicap when dealing with IC/BPS.

- you have to be more detailed about phenotyping. It is true that it is a great challenge for diagnosis and treatment of IC/BPS, but how phenotypes could come out through a national database? If it is just a survey hypothesis, you have to document it.

Author Response

This is an excellent article. I have only some minor comments before it could be published.

Reply: thank you for these kind words

- although the recruitment has been pooled by a national database, it remains unclear if the diagnosis of IC/BPS really meets the ESSIC criteria. You probably have to document the limitations occurring due to this common handicap when dealing with IC/BPS.

Reply: this is entirely true since this is a fully patient reported dataset. We added in the discussion this as limitation: Patients provided information concerning their diagnosis themselves and this data could not be corroborated with actual patient clinical data due to the study methodology.

- you have to be more detailed about phenotyping. It is true that it is a great challenge for diagnosis and treatment of IC/BPS, but how phenotypes could come out through a national database? If it is just a survey hypothesis, you have to document it.

Reply: thanks for this comment: we asked patients on the basis of the ESSIC criteria if they have had Hunner’s lesions, glomerulations or no bladder abnormalities. Due to the nature of the study we could not verify this in clinical patient files. Nonetheless, we did find a small difference in symptom severity between the HL and non-HL groups, which does underscore that phenotyping is relevant. We added in the discussion: Patients provided information concerning their diagnosis themselves and this data could not be corroborated with actual patient clinical data due to the applied study methodology. This has to be taken into account when interpreting the reported differences in symptom severity between HL patients and non-HL patients.

Reviewer 3 Report

Comments and Suggestions for Authors

This study addresses a crucial evidence gap in the field of interstitial cystitis/bladder pain syndrome (IC/BPS) by focusing on the quality of life (QoL) of affected individuals. The authors highlight the limitations of existing guidelines, emphasizing the scarcity and outdated nature of studies, which may compromise the accuracy of current recommendations. The investigation into patient perspectives on QoL and their evaluation of various therapies adds valuable insights, especially in the context of the recommended multimodal approach. The utilization of a quantitative retrospective database study, drawing from a substantial pool of patient survey data, enhances the robustness of the findings. The authors effectively demonstrate that IC/BPS significantly impacts QoL, as evidenced by self-reported data and the multifaceted effects on EQ-5D 5L domains. Notably, the study unveils that delayed diagnosis does not necessarily exacerbate QoL impairment, although gender differences are observed. The identification of a wide array of treatments, including alternative therapies, provides a comprehensive overview. The acknowledgment of some treatments with doubtful results and high dropout rates underscores the need for a critical evaluation of therapeutic approaches.

Nevertheless, some important points have to be clarified or fixed before positive actions with regards to publication can be taken.

The title accurately reflects the content. Even though, in general, the abstract presents an adequate synopsis of the paper, it requires major editing. Ensure the abstract is free from grammatical errors and maintains a consistent scientific style. This will contribute to the professionalism of the manuscript and help convey your message more effectively. The authors should carefully proof-read the text with a writing coach or copyeditor to eliminate grammatical errors and improve sentence construction, word choice and clarity.

Clarify Questionnaire Timing: Specify the time frame in which participants received their IC/BPS diagnosis relative to completing the questionnaire. This information is crucial for contextualizing responses and understanding the relevance of the data collected.

Acknowledge Potential Bias: Recognize any potential biases in the data collection process, which may introduce selection bias.

Provide additional details regarding the selection criteria, survey administration, and any steps taken to ensure data quality. This proactive approach demonstrates a commitment to transparency and strengthens the overall methodology section.

Response rate: To enhance the transparency of participant engagement, I recommend including the response rate in the manuscript. This information is crucial for understanding the representativeness of the sample and the generalizability of the findings. Please include the total number of individuals invited or eligible to participate, along with the number of respondents, and express the response rate as a percentage. This addition will strengthen the methodology section and contribute to a more thorough interpretation of the study results

Comments on the Quality of English Language

There is a certain level of inconsistency observed regarding the language quality, particularly in the abstract and introduction sections. Editing is needed to ensure clarity, coherence, and grammatical accuracy throughout these critical sections. Review and refinement of language usage and sentence structure would significantly enhance the overall readability and professionalism of the manuscript

Author Response

This study addresses a crucial evidence gap in the field of interstitial cystitis/bladder pain syndrome (IC/BPS) by focusing on the quality of life (QoL) of affected individuals. The authors highlight the limitations of existing guidelines, emphasizing the scarcity and outdated nature of studies, which may compromise the accuracy of current recommendations. The investigation into patient perspectives on QoL and their evaluation of various therapies adds valuable insights, especially in the context of the recommended multimodal approach. The utilization of a quantitative retrospective database study, drawing from a substantial pool of patient survey data, enhances the robustness of the findings. The authors effectively demonstrate that IC/BPS significantly impacts QoL, as evidenced by self-reported data and the multifaceted effects on EQ-5D 5L domains. Notably, the study unveils that delayed diagnosis does not necessarily exacerbate QoL impairment, although gender differences are observed. The identification of a wide array of treatments, including alternative therapies, provides a comprehensive overview. The acknowledgment of some treatments with doubtful results and high dropout rates underscores the need for a critical evaluation of therapeutic approaches.

Reply: Thanks for reviewing the manuscript. We have adapted it according to your suggestions

Nevertheless, some important points have to be clarified or fixed before positive actions with regards to publication can be taken.

The title accurately reflects the content. Even though, in general, the abstract presents an adequate synopsis of the paper, it requires major editing. Ensure the abstract is free from grammatical errors and maintains a consistent scientific style. This will contribute to the professionalism of the manuscript and help convey your message more effectively. The authors should carefully proof-read the text with a writing coach or copyeditor to eliminate grammatical errors and improve sentence construction, word choice and clarity.

Reply: we have extensively reviewed and improved (especially the Abstract) the English of this manuscript. We hope this will make in more clear and readable.

Clarify Questionnaire Timing: Specify the time frame in which participants received their IC/BPS diagnosis relative to completing the questionnaire. This information is crucial for contextualizing responses and understanding the relevance of the data collected.

Reply: We have evaluatied this, We added in the text in results section (sentence 138): over 60% of patients received their diagnosis within the last 10 yrs at the time of filling in the survey, with a smaller group receiving diagnosis earlier.

Acknowledge Potential Bias: Recognize any potential biases in the data collection process, which may introduce selection bias.

Reply: we have exended the section on potential limitations and biases of this study: There are some limitations to this study. One potential limitation of this study lies in its retrospective nature, which could potentially increase the likelihood of recall bias. No comparator group of age matched healthy volunteers was used in this study and the male group was still comparatively small. The study was conducted with patient data from one single Western European country and healthcare management for IC/BPS patients can differ between countries. Patients provided information concerning their diagnosis themselves and this data could not be corroborated with actual patient clinical data due to the applied study methodology. This has to be taken into account when interpreting the reported differences in symptom severity between HL patients and non-HL patients.

For patient reported satisfaction with treatment, there were limitations as to how to interpret these. Not all therapies can be directly compared to each other since they may have been offered to patients at different stages with periods of different symptom severity / burden. Also the perception of treatment efficacy could be mediated by other treatments that were given at the same time. The survey was made accessible to individuals through the Dutch IC/BPS patient association. Nonetheless, since participation was voluntary, it may be susceptible to response bias. This may be reflected in the relatively high number of HL patients in this dataset.

Provide additional details regarding the selection criteria, survey administration, and any steps taken to ensure data quality. This proactive approach demonstrates a commitment to transparency and strengthens the overall methodology section.

Reply: we added in M & M: The online survey was performed using SurveyMonkey and data was fully based on anonymity for the participants. The Dutch patient association ICP advertised this survey on their website and social media platform. Data was stored electronic and secure and only accessible by researchers.

Response rate: To enhance the transparency of participant engagement, I recommend including the response rate in the manuscript. This information is crucial for understanding the representativeness of the sample and the generalizability of the findings. Please include the total number of individuals invited or eligible to participate, along with the number of respondents, and express the response rate as a percentage. This addition will strengthen the methodology section and contribute to a more thorough interpretation of the study results

Reply: the survey was conducted on an open platform, and advertised by the Dutch Patient association for BPS/IC (ICP). The survey was not sent directly to participants, therefore we do not have information response rates.

Comments on the Quality of English Language

There is a certain level of inconsistency observed regarding the language quality, particularly in the abstract and introduction sections. Editing is needed to ensure clarity, coherence, and grammatical accuracy throughout these critical sections. Review and refinement of language usage and sentence structure would significantly enhance the overall readability and professionalism of the manuscript

Reply: we extensively edited the abstract and main manuscript on English language.

Round 2

Reviewer 1 Report

Comments and Suggestions for Authors

The revisions made by the authors significantly enhanced the clarity of the manuscript. However, I would like to address two additional comments/suggestions:

1. In the methods section, the authors indicated that the analysis included "only adult patients who had received the IC/BPS diagnosis at the time of the questionnaires" (lines 88 and 89). Yet, in the results section, there seems to be a contradiction as the authors state that "Over 60% of patients received their diagnosis within the last 10 years at the time of filling in the survey, with a smaller group receiving the diagnosis >10 years ago" (lines 134-136). It is crucial for the authors to provide clarification in the methods section to align with the information presented in the results section.

2. In response to a previous comment, the authors included additional information about the EQ-5D 5L questionnaire used to assess quality of life and the O’Leary Sant Interstitial Cystitis Symptom Index and Problem Index (ICSI/PI) in the titles of tables. While this information aids in the interpretation of results, I would recommend incorporating it into footnotes rather than the titles for better organization and clarity.

Author Response

Remark: In the methods section, the authors indicated that the analysis included "only adult patients who had received the IC/BPS diagnosis at the time of the questionnaires" (lines 88 and 89). Yet, in the results section, there seems to be a contradiction as the authors state that "Over 60% of patients received their diagnosis within the last 10 years at the time of filling in the survey, with a smaller group receiving the diagnosis >10 years ago" (lines 134-136). It is crucial for the authors to provide clarification in the methods section to align with the information presented in the results section.

Reply: Thanks for this valid remark: we amended the M&M section: and changed it into (line 95-96): 'For the analysis, only adult patients who reported to have been diagnosed with IC/BPS and who reported to still have IC/BPS at the time of the questionnaire were included.' This aligns it more clear with outcomes of the Results section.

For the 2nd remark; In response to a previous comment, the authors included additional information about the EQ-5D 5L questionnaire used to assess quality of life and the O’Leary Sant Interstitial Cystitis Symptom Index and Problem Index (ICSI/PI) in the titles of tables. While this information aids in the interpretation of results, I would recommend incorporating it into footnotes rather than the titles for better organization and clarity.

Reply: we redirected the information on ICSI/PI en EQ-5D-5L to the footnotes of the tables.